# Applications of Single-Cell Sequencing Technology to the Enteric Nervous System

**DOI:** 10.3390/biom12030452

**Published:** 2022-03-15

**Authors:** Richard A. Guyer, Jessica L. Mueller, Allan M. Goldstein

**Affiliations:** Department of Pediatric Surgery, Massachusetts General Hospital, Boston, MA 20114, USA; rguyer@mgh.harvard.edu (R.A.G.); jmueller2@mgh.harvard.edu (J.L.M.)

**Keywords:** enteric nervous system, single cell RNA sequencing, single cell epigenetics, neural crest, neurogenesis

## Abstract

With recent technical advances and diminishing sequencing costs, single-cell sequencing modalities have become commonplace. These tools permit analysis of RNA expression, DNA sequence, chromatin structure, and cell surface antigens at single-cell resolution. Simultaneous measurement of numerous parameters can resolve populations including rare cells, thus revealing cellular diversity within organs and permitting lineage reconstruction in developing tissues. Application of these methods to the enteric nervous system has yielded a wealth of data and biological insights. We review recent papers applying single-cell sequencing tools to the nascent neural crest and to the developing and mature enteric nervous system. These studies have shown significant diversity of enteric neurons and glia, suggested paradigms for neuronal specification, and revealed signaling pathways active during development. As technology evolves and multiome techniques combining two or more of transcriptomic, genomic, epigenetic, and proteomic data become prominent, we anticipate these modalities will become commonplace in ENS research and may find a role in diagnostic testing and personalized therapeutics.

## 1. Introduction

Cellular diversity has long been known to exist in the enteric nervous system (ENS), with enteric neuronal subclasses traditionally defined based on their morphology, anatomic location of their cell bodies and projections, and neurochemical profile. Immunohistochemistry has been most commonly used to identify the panel of neurotransmitters expressed by different neurons as a way of classifying them into functional subtypes [1,2]. The identification of subpopulations of enteric glia has been similarly based on morphologic differences, cellular location, and immunophenotype [3,4]. These classification systems have identified neuronal and glial diversity and offered invaluable information regarding their potential function and interrelationships among these cells. The recent ability to determine the transcriptional profile of enteric neurons and glia at the single-cell level has significantly enhanced and refined our understanding of ENS anatomy, biology, and pathophysiology.

Tools for analysis at single-cell resolution, such as high-resolution microscopy and flow cytometry, have existed for decades. However, these traditional methods are limited to measuring a small number of parameters in a single assay. Other techniques, such as bulk RNA sequencing and mass spectrometry-based proteomics, can assess many parameters in a single experiment but measure average signal from a large number of cells. Highly parameterized studies on bulk populations have generated important biological insights, such as identifying conserved transcription factors in mouse and human enteric neuron development [5], but these methods cannot resolve heterogeneity and are poorly suited to studying rare cell populations. Such limitations have spurred development of tools for high-parameter measurement of genomic, transcriptomic, epigenetic, and protein variables in single cells made feasible by next-generation sequencing platforms and cellular barcoding technology. These technologies have undergone rapid evolution since their introduction, with significant improvements in the number of unique features identifiable in each cell and recent introduction of “multiome” approaches capable of simultaneously measuring two or more feature types (RNA, chromatin accessibility, protein expression, etc). These tools have significantly improved our understanding of cellular diversity and function within numerous tissues, including the ENS.

In this review, we summarize current single-cell sequencing modalities, consider insights from single-cell sequencing of ENS cells, and comment on ways these technologies may impact the field in coming years. Because the ENS is neural crest-derived, we include papers that discuss the early specification and differentiation of the neural crest lineage. To our knowledge, no papers in the peer reviewed literature have yet applied single-cell technologies other than RNA sequencing to the ENS. However, we expect that single-cell measurement of other variables will be performed on ENS populations in the near future, and it is important for ENS researchers to be aware of these technologies, so they are discussed here. While exciting advances are occurring in mass spectrometry-based methods for single-cell proteomics [6,7], such technology remains nascent and is not reviewed here.

## 2. Overview of Single-Cell Sequencing Technologies

Single-cell sequencing technologies can be classified based on their approach to isolating single cells or, in some cases, single nuclei, and by the feature or features they measure. From a technical perspective, cells can be isolated by sorting or limiting dilution into single wells, followed by amplification and library generation; such methods include Smart-Seq [8,9] and CEL-Seq [10,11]. When combined with high-depth sequencing, these approaches allow for the reliable detection of large numbers of transcripts, including genes with very low expression, and permit the application of complex analyses, such as detection of splice variants. However, the limitations imposed by multiwell plates prevent these methods from assaying large numbers of cells. Droplet-based approaches, such as Drop-Seq [12] and the 10× Chromium system [13], use an excess of individually-barcoded gel beads and microfluidic technology to isolate individual cells or nuclei prior to library construction. Droplet technology permits far more cells to be assayed in a single experiment, although the trade-off between cell number and reads per cell means these methods typically do not approximate whole transcriptome coverage. Finally, combinatorial barcoding methods such as sci-RNA-seq [14] do not isolate individual cells prior to library formation, but rather use several cycles of barcoding pools of cells. Between cycles of barcoding, cells are mixed and randomly assigned to a new barcode pool. The chances of multiple cells receiving the same set of barcodes is quite low, thus permitting low-cost library generation from large numbers of cells. When choosing from among these techniques, investigators should know there is currently no preferred approach. The choice of technique is largely based on the biological question being addressed, investigator preference and expertise, and available resources. Most studies examining the ENS thus far have used droplet-based approaches (Table 1), but this likely reflects widespread availability of these systems, and there is no theoretical reason why other approaches could not be used.

Regardless of the method used for applying unique barcodes to individual cells, sequencing libraries are constructed from pools of many single cells (or nuclei), and sample barcoding can allow cells from several replicates or unrelated experiments to be combined into a single library. The libraries are sequenced on next-generation platforms. Software pipelines, such as 10× Genomics’s Cell Ranger suite [13], have been developed for demultiplexing reads and delivering cell-feature count matrices. Similarly, numerous open-source software packages have been developed for analyzing and visualizing these data. Two of the most popular are the R-based Seurat [30] and the Python-based SCANPY [31]. Extensions to these packages for processing epigenetic and multiome data are available [32,33]. Probably due to the less widespread use of single-cell genomic and proteomic methods, analysis of these data typically relies on custom software developed by individual laboratories rather than widely utilized packages.

Heterogeneous gene expression within tissues, even among superficially indistinguishable cells, has been appreciated for some time. Consequently, attempts at single-cell transcript sequencing date back three decades, when Eberwine and colleagues described a method for generating cDNA libraries from the whole transcriptome of individual hippocampal rat neurons [34]. This early report included the observation that morphologically-similar cells vary widely in their expression of particular genes. Unfortunately, this method was technically challenging, and global assessment of genes was not possible with technology available at the time. With the advent of next-generation sequencing platforms capable of generating large numbers of reads at low cost [35], unbiased assessment of individual cells’ whole transcriptome became feasible. Since the first report of using next-generation technology to sequence the entire transcriptome of a single mouse blastomere [36] there have been significant advances, such that tens of thousands of cells can now routinely be surveyed in a single experiment [37]. Single-cell RNA sequencing is now a commonplace tool used by biologists [38].

It has long been appreciated that unicellular organisms of the same species, and genome sequences among the many cells of multicellular organisms, are not necessarily identical [39]. This is most strikingly true in cancer [40,41], but heterogeneity inevitably increases over time within multicellular organisms due to accumulating mutations [42,43]. Driven by a desire to understand genomic heterogeneity and to use clonal variations to map cellular hierarchies, scientists have developed and improved numerous approaches to single-cell genome sequencing over the past decade. The first such report came in 2011, when Navin et al. flow-sorted 200 total individual nuclei from two breast tumors followed by whole-genome amplification and sequencing [44], thus permitting reconstruction of tumor evolution and the emergence of metastatic clones. Similar analyses have revealed significant clonal variation in the central nervous system, where emergence of aberrant clones may be an important cause of neurologic diseases [45,46]. We expect that single-cell genomic analysis of the ENS would be a similarly rich source of biological insights, but to date no such efforts have been reported.

There has been rapid growth in the techniques for analyzing epigenetic variables in single cells. Such variables include chromatin accessibility, DNA methylation, histone modifications, and transcription factor binding. While numerous methods for assessing DNA methylation have been reported [47,48,49], no published studies have yet applied these techniques to the ENS. Similarly, assaying chromatin accessibility via the assay for transposase-accessible chromatin (ATAC) is a powerful tool for understanding gene regulation, identifying active chromatin regions, and resolving heterogeneous cell populations. Though initially developed for studying bulk populations [50], protocols have now been developed for single-nucleus ATAC [51,52]. The most widespread technique utilizes the popular 10× Genomics platform, making this tool accessible to most laboratories. Although chromatin accessibility gives valuable information regarding a cell’s transcriptional potential and allows for the resolution of complex populations, additional insights are possible from studying histone modifications and binding patterns of specific transcription factors. This motivated the adaption of single-nucleus ATAC technologies to accommodate CUT&Tag technology [53], whereby antibody-bound Tn5 transposase is used to isolate genomic regions bound by specific proteins. Histone and transcription factor profiling via snCUT&Tag has been described using both the 10× Genomics droplet-based systems [54] and via combinatorial barcoding approaches [55]. As with DNA methylation, no peer-reviewed studies have yet been published applying snATAC or snCUT&Tag to the ENS.

The explosion in single-cell sequencing technologies over the past decade has naturally caused interest in the simultaneous assay of multiple variables in individual cells. Multiome technologies that simultaneously measure parameters from two or more biological traits are increasingly common. Such measures are particularly valuable when one variable, such as expression of a known set of genes, is considered a gold standard for distinguishing populations and another, such as the histone methylation pattern, is expected to vary between populations but would not suffice alone to determine cell identity confidently. Multiome data can also provide snapshots of the relationship between chromatin structure and gene expression across diverse cell populations, allowing investigators to generate testable hypotheses regarding transcriptional regulation and identify candidate transcription factors controlling cell programs. Multiome approaches have been described in single cells for joint RNA and ATAC [56], T cell receptor sequencing and RNA [57], histone modifications and RNA [55], and cell-surface antigen expression and RNA [58]. These technologies have not yet been applied to the ENS in the peer-reviewed literature.

All the single-cell sequencing tools discussed thus far require dissociation of cells from their native tissue environment. This eliminates spatial information and prevents these tools from understanding how complex cell types are arranged functionally within organs. Techniques for in situ RNA and genome sequencing have been described to overcome these shortfalls [59,60]. These methods rely on overlaying tissue sections on slides containing bead arrays. Each bead on the slide contains Illumina adaptors and unique barcode sequences which capture RNA or DNA from the tissue region placed directly over the bead. While this approach generally does not obtain as many unique reads per region as single-cell sequencing, the data obtained does suffice to reconstruct tissue architecture. Unlike single-cell sequencing, it is impossible to ensure that each bead is only measuring one cell. Resolution is limited by bead size, currently ten microns across, making this tool poorly suited to the in situ ENS in which ganglia are quite small and neurons and glia are very tightly apposed to one another. However, given the close approximation of neurons and glia within enteric ganglia and the challenges of dissociating these cells, in situ sequencing may prove extremely valuable to the ENS community if technical improvements result in improved resolution.

## 3. Review of Current Single-Cell Sequencing Analyses of the Enteric Nervous System

As single-cell sequencing modalities have matured and become widely available, scientists have used these platforms to study development, diversity, and function in the ENS. The first application to the ENS was reported in 2017 by Lasrado and colleagues, who used a fluidic system to capture 192 single cells marked by a tdTomato reporter driven by Sox10 expression from mouse embryos at E12.5 [15]. After filtering for high-quality cells and removing potential doublets, they analyzed 120 ENS progenitors. Though far smaller than subsequent studies, this was a significant technical achievement at the time and provided valuable insights. Of the 120 cells examined, the great majority fell into one of three groups: (1) cells expressing glial and progenitor marker genes, such as *Sox10*, *Erbb3*, and *Plp1*, without any neuronal gene expression; (2) cells co-expressing glial and progenitor genes along with neuronal genes, such as *Tubb3* and *Elavl4*; and (3) cells expressing only neuronal genes. A small group with low expression of all ENS genes was also identified, which are possibly contaminant cells. Using principal component analysis, the authors demonstrated these three groups of cells exist along a continuum between the purely glial/progenitor cells and the purely neuronal cells, and they identified Ret as a possible driver of neuronal differentiation based on its high level of expression in both neuronal cells and cells with a mixed phenotype.

A short time later, a group led by Allon Klein published two large-scale studies of vertebrate model systems at various embryonic stages. The authors used a droplet-based approach to sequence approximately 92,000 cells from *Danio rerio* embryos and approximately 137,000 cells from *Xenopus tropicalis* embryos, in both cases pooling cells from sequential time points [16,17]. Both of these datasets generated high-quality developmental atlases of diverse tissues and organs. The authors developed a method for two-dimensional visualization of the data that places cells on a branching network with cells from later time points further along the branches. This approach revealed dynamic gene expression changes occurring as cells progress from pluri- and multipotent progenitors, to lineage committed progenitors, and ultimately to differentiated states. While both of these studies captured a significant number of neural crest lineage cells, the authors only performed an in-depth analysis of the neural crest lineage in the *Xenopus* model. By examining the genes expressed at stages between the blastula to the early neural crest, they found no evidence of a persistent transcriptional program beyond the blastula. While conceding that functional pluripotency could be maintained by epigenetic mechanisms or other, non-transcriptional regulatory mechanisms, they argued that it is most likely that the neural ectoderm reacquires multipotency during early neurulation.

In a tremendous undertaking, Zeisel et al. reported single cell sequencing of 509,876 murine nervous system cells [18], 492,949 of which were of sufficient quality for inclusion in their computational analysis. This included over 10,000 ENS cells isolated by sorting fluorescent cells from the myenteric plexus layer of the small intestine from *Wnt1Cre;R26Tomato* transgenic mice around postnatal day 21. Of the ENS cells, over 90% were enteric glial cells. While in-depth analysis of the ENS was not reported, they did note that their clustering algorithm suggested seven distinct subtypes of enteric glial cells, including actively proliferating cells. Nine subtypes of enteric neurons were identified, including nitrergic neurons expressing *Nos1* and *Chat*-expressing cholinergic neurons. Despite the limited analysis of the ENS undertaken by the authors, this paper hinted at significant diversity among both neurons and glia in the ENS. The dataset has been a valuable resource for the ENS community.

Without directly addressing the developing ENS, Soldatov and colleagues combined scRNAseq with in situ hybridization to identify cell fate decision points during neural crest differentiation [20]. They performed single cell sequencing on cranial neural crest cells at E8.5, trunk and vagal neural crest cells at E9.5 and E10.5, and neural crest cells migrating within the hindlimb and tail at E10.5. The differentiation tree suggested by single cell sequencing implies a series of binary cell fate decisions, with an early split into sensory neurons versus other potential neural crest fates, followed by the splitting of autonomic neuronal and mesenchymal lineages. They validated these results with lineage tracing using markers of sensory neurons (*Phox2b*), autonomic neurons (*Neurog2*), and mesenchymal cells (*Prrx1*). This model of sequential binary fate decisions correlates with the results of Morarach et al. [25] (discussed below) regarding neuronal differentiation in the embryonic ENS. Intriguingly, Zeisel and colleagues [18] reported a population of fibroblast-like cells among their tdTomato-positive ENS glial cells, suggesting that these cells are neural-crest derived. The presence of neural crest-derived fibroblast-like cells within the gut has yet to be validated by in vivo hybridization or immunostaining, so doublet contamination within the Zeisel dataset remains a possibility. Nevertheless, these data raise the possibility that neural crest cells are precursors for more than neurons and glia in the gastrointestinal tract, which would alter the existing paradigm that the only neural-crest derived cells in the gut are neurons and glial cells.

Late in 2019, Lau and colleagues reported the first use of single-cell technology to directly test a biological hypothesis regarding ENS development [21]. This differed from prior studies that had used single-cell technology to describe and characterize ENS and early neural crest populations. Lau et al. used the 10× Genomics droplet-based scRNAseq system to compare transcriptional trajectories during enteric neuronal differentiation of human induced pluripotent stem cells and in vivo mouse ENS formation. They found compelling evidence that these model systems closely resemble one another, suggesting that the process of enteric neurogenesis is highly conserved among mammals and that experiments utilizing mouse models are likely to yield insights relevant to human biology. They then used a single-cell approach to understand hedgehog signaling dynamics during enteric neurogenesis. Using a fluorescent reporter system, they isolated cells with high and low levels of hedgehog activation and performed scRNAseq. Cells from both populations co-clustered with cells at all stages of neuronal differentiation, which the authors interpreted as meaning that hedgehog signaling activity is dynamically changing throughout neurogenesis. They tested this observation via numerous approaches, including scRNAseq of human induced pluripotent stem cells undergoing ENS differentiation in the presence or absence of a hedgehog agonist, and showed that hedgehog agonism accelerates neuronal differentiation.

In an effort to generate a comprehensive single-cell atlas of the mature mouse and human ENS, Drokhlyansky and colleagues devised numerous methods to isolate and sequence enteric neurons and glia [22]. First, they developed a novel method for isolating intact nuclei and ribosomes from single cells followed by droplet-based single-nucleus RNA sequencing with the 10× Genomics system, which they called RAISIN RNA-seq. When they applied RAISIN RNA-seq to ENS cells from the mouse intestine they identified 21 neuronal subtypes and three clusters of glia. This glial diversity is significantly lower than previously reported by Zeisel et al. [18]. Because they sequenced a relatively small number of neurons per animal, it is unclear whether all of the neuronal clusters represent distinct biological entities or are attributable to batch effects and inter-specimen variation. Isolating ENS cells from human bowel is a significant technical challenge, since fluorescent reporters are not available to enrich the relatively rare ENS populations. They overcame this obstacle via a technique they called “mining rare cells sequencing,” or MIRACL-seq. MIRACL-seq involves intentionally overloading a droplet-based system despite the high doublet rate. Doublets involving two cells from a rare population are very unlikely, meaning cells only exhibiting a gene expression pattern characteristic of a rare population can be assumed to be singlets and used for analysis. This approach is extremely resource-intensive, as 704,314 nuclei isolated from colon specimens were sequenced to obtain single-cell data from just 1445 neurons and 6054 glial cells. They identified 14 clusters of enteric neurons in the adult human colon, including groups of putative motor neurons, sensory neurons, and interneurons. Six clusters of human glia were found, although three of these were unique to individual tissue donors, suggesting significant batch effects. Using the data obtained from these efforts, the authors examined gene expression differences in the proximal versus distal mouse colon, as well as circadian effects. Although they did find differences when stratifying by these variables, the size of gene expression changes was small, there was considerable variance in the data, and batch effects due to the small number of cells from each donor mouse may have confounded the results. They also compared ENS gene expression programs in the mouse and human colon and found them to be quite similar, thus recapitulating Lau et al.’s prior report that development of the mouse and human ENS are highly conserved processes [21]. This work was a significant technical achievement, and the resulting data are valuable, but the enormous resources RAISIN RNA-seq and MIRACL-seq require render these methods unlikely candidates for widespread adoption.

In the most extensive study of enteric neurons to date, Morarach and colleagues used a *Baf53b-Cre;R26R-Tomato* system to sort neurons from the gut of mice on postnatal day of life 21, as well as a *Wnt1-Cre;R26R-Tomato* system to isolate all neural-crest derived cells from the small intestine of embryonic mice at E15.5 and E18.5 [25]. After isolating cells, the 10× Genomics system was used for scRNAseq. They obtained high-quality data from 4892 P21 neurons, the analysis of which revealed 11 clusters of neurons, including excitatory and inhibitory motor neurons, sensory neurons, interneurons, and one cluster whose function could not be inferred from its gene expression pattern. They identified transcription factors likely to determine each neuronal fate and, in a laborious undertaking, performed extensive immunofluorescence to validate expression of markers for each neuronal subtype in gut tissue. Using the data obtained from E15.5 and E18.5 cells, the authors mapped the differentiation of neural crest cells into each subtype of neuron and identified factors likely to drive cell fate decisions. Interestingly, their data strongly suggests plasticity between neuronal subtypes during prenatal development, with early emergence of only two types of neurons—excitatory motor neurons marked by *Bnc2* and inhibitory motor neurons marked by *Etv1*—followed by transition from these two initial classes into other neuronal types. This initial binary fate choice followed by further differentiation is reminiscent of the sequential binary fate choices observed by Sodatov et al. [20] in the non-ENS peripheral nervous system. Whether plasticity among neuronal subtypes persists in the postnatal ENS is likely to be an area of intensive investigation in the coming years.

Elmentaite and colleagues have published two large-scale scRNAseq analyses of the human intestine. The first of these papers sequenced over 62,000 cells from the intestine of human fetuses between 6 and 11 weeks post-conception, as well as over 11,000 cells from mucosal biopsy samples of children aged 4–12 years [23]. Although this first study captured a significant number of ENS progenitors and nascent neurons, as well as a small number of postnatal glial cells (which are presumably submucosal glia rather than myenteric plexus glia), no significant analysis of these cells was reported. Subsequently, they profiled over 400,000 human intestinal cells at locations ranging from the duodenum to the rectum, including both human fetuses 12–17 weeks post-conception and adults [26]. They integrated this additional data with their earlier datasets for analysis, including reconstruction of cell lineages in the developing ENS. This data showed remarkable similarity to Morarach et al.’s mouse data [16], including initial branching of ENS progenitors into *BNC2*- and *ETV1*-expressing neurons. These studies provide a tremendous amount of hypothesis-generating data for investigators, as well as offering further evidence that mammalian ENS development is highly conserved.

To identify functional regulators of the postnatal ENS, Wright and colleagues used both snRNAseq and scRNAseq of adult mouse colon, adult human colon, and embryonic day 17.5 mouse ENS cells from both the large and small intestine [27]. They identified seven types of enteric neurons in the adult mouse and eight subtypes in E17.5 mice and found hundreds of neurotransmitters, neurotransmitter receptors, ion channels, messenger RNA regulatory genes, and signaling molecules to be differentially expressed across these clusters. They found adult neuron subtype transcription factor patterns to be established by E17.5, including factors previously identified by Morarach et al., such as *Casz1*, *Bnc2*, *Etv1*, *Pbx3*, and *Tbx3* [25]. Conditional knockout of *Tbx3* reduced Nos1+ neuron density by 30%, thus validating a role for this factor in specifying nitrergic neurons. Additionally, the authors identified that *Gfra1* and *Gfra2*, which encode the receptors for GDNF and NRTN, respectively, are differentially expressed across neuron subtypes, with *Gfra1* expressed primarily in inhibitory *Nos1*/*Vip*/*Gal*+ neurons, and *Gfra2* expressed in cholinergic *Chat*+ neurons. Coupled with calcium imaging, the authors found GDNF and NRTN to influence the activity of ~50% of myenteric neurons, and using functional studies, they reported that GDNF, but not NRTN, acutely enhances colonic contractility. The study did not identify or validate as many subtypes of postnatal neurons as Morarach et al. [25], but nevertheless it provided further evidence of scRNAseq’s utility to characterize diverse ENS populations and identify important regulatory factors.

May-Zhang et al. used several approaches to transcriptionally profile duodenal, ileal, and colonic enteric neurons from both adult human gut tissue and mice [28]. Human myenteric ganglia were isolated by laser capture microdissection and subjected to bulk RNA sequencing, while mouse enteric neurons were isolated by FACS sorting utilizing a *Phox2b* promoter-driven Histone 2B-CFP fusion transgene. In both species, there was regionally-restricted expression of certain genes, such as *CCKAR*, *POU3F3*, and *HOX3A*, suggesting location-specific ENS functions. Markers of neuronal subtypes were largely conserved across species. One important difference was identified in the intrinsic primary afferent neurons (IPANs). *Nmu* and *Klhl1* were present in murine IPANs, whereas in humans, *NMU* labeled IPANs and *KLHL1* labeled a distinct population. This study was published around the same time as the paper by Morarach et al. [25], so neither study directly compared the neuronal clusters identified by each.

The Koohy and Simmons groups collaborated to perform scRNAseq and spatial RNA sequencing of full-thickness intestinal samples from 8- to 22-week-old human embryos [29]. They catalogued over 76,000 cells, including epithelial cells, fibroblast, endothelial cells, pericytes, neurons, glia, smooth muscle cells, mesothelium, myofibroblasts, and immune cells. The authors identified putative regulatory genes influencing cell fate for each cell type. They found 12 cell clusters within the ENS lineage, including five clusters they label as glial and seven as neuronal. Interestingly, these cells organize into a branching lineage structure that closely resembles the differentiation trajectory reported by Morarach et al. [25], although the authors did not directly compare their data with prior reports.

Baghdadi et al. paired scRNAseq techniques in mice and humans with in vivo ablation of specific enteric glial cell (EGC) populations to define the heterogeneity of glial cells and identify their role in intestinal stem cell homeostasis [61]. This paper differed from prior scRNAseq studies of the ENS in its focus on biological mechanisms rather than merely characterizing cellular diversity. After initially noting that ablation of Gfap+, but not Plp1+, EGCs reduced the expression of numerous stem cell markers and the frequency of intestinal stem cells in crypts, they examined a previously-published scRNAseq dataset of cells from the lamina propria of the murine colon [24] which included some glial cells. Within this dataset they identified three types of glial cells based on PLP1 and GFAP expression: GFAP^High^/PLP1^Low^, GFAP^Low^/PLP1^High^, and GFAP^Mid^/PLP1^Low^. Simultaneous ablation of both Plp1+ and Gfap+ glial cells in mice caused catastrophic intestinal mucosal breakdown, resulting in death within several days, indicating that glial cells play an essential role in epithelial homeostasis. They found that Gfap+ glial cells were significantly increased in mice following radiation- or cytokine-induced inflammatory injury, and that these Gfap+ cells facilitate epithelial repair by providing Wnt ligands to support the epithelial stem cell niche. They correlated these results with a previously published scRNAseq dataset of mesenchyme from both healthy and ulcerative colitis human colon tissue [19], finding that a glial subpopulation that resembles the GFAP^High^/PLP1^Low^ murine cells is increased in UC tissue. While this study establishes a functional role for glial cells in intestinal epithelial homeostasis, it does not delineate whether the myenteric plexus glia and submucosal glia have distinct effects on the epithelium. Notably, neither scRNAseq dataset used was specifically enriched for ENS cells and the murine dataset only included cells from submucosal tissue, so the results may not be representative of myenteric plexus glia.

## 4. Looking toward the Future

Over the past five years there have been numerous ENS applications of scRNAseq, resulting in important biological insights. There is significant transcriptional diversity among ENS neurons and glial cells, the functional implications of which will require further work to unravel. In the case of neurons, transcriptional diversity has been carefully validated by immunostaining [25]. Less effort has been expended to validate glial subpopulations in studies published to date, so it is not certain whether the glial diversity reported reflects biologically relevant populations. With the exception of Lau et al.’s application of scRNAseq to differentiating induced pluripotent stem cells in the presence or absence of hedgehog agonists [21], most single-cell data collected thus far in the ENS has been descriptive. These descriptive studies have advanced the field tremendously and provided rich resources that will aid ENS investigators for many years to come. Nevertheless, we anticipate a transition in the near future from predominantly descriptive studies to using single-cell sequencing tools to test specific hypotheses.

Notably, the numerous studies published to date have reported varying numbers of neuronal and glial populations. The numbers of glial clusters identified range from three types of mouse glia found by Drokhylansky et al. [22] to seven glial subtypes identified in mice by Zeisel et al. [18]. Drokhylansky et al. also reported six transcriptionally distinct glial clusters in human tissue, but three of these clusters were specific to individual tissue donors [22]. Neuronal numbers are similarly variable, with numbers ranging from 21 neuronal types identified in mice by Drokhylansky et al. [22] to the seven types of neurons identified by Wright et al. [27] and Fawkner-Corbett et al. [29]. Human studies have found less neuronal diversity than mouse studies, which we speculate is due to the greater challenges of isolating high-quality cells from human tissue. Factors that might impact intraspecies variation in cluster numbers include the method of isolating cells, the scRNAseq protocol use, rigor of filtering low-quality cells from datasets, and the algorithms and parameters used for clustering. Consensus within the field regarding markers for various cell types and widely accepted parameters for analyzing datasets may help resolve these discrepancies. Studies that use predetermined parameters to integrate and analyze multiple datasets may also produce valuable insights.

Exciting work has been done in recent years to develop single-cell sequencing methods for interrogating epigenetic traits such as chromatin accessibility and histone modifications [54,55,62]. However, no studies have applied these methods to the ENS. We anticipate that single-cell epigenetic tools, especially when jointly performed with scRNAseq, will offer tremendous insights into the developmental lineages and postnatal plasticity of both neurons and glial cells. Epigenetic mechanisms likely underlie the capacity of glial cells to generate neurons under appropriate conditions [63,64,65]. Similarly, Morarach et al. suggested that identity switching is routine during developmental specification of enteric neurons [25]. Whether such plasticity among neurons persists into adult life is currently unknown, but we speculate that if neurons do remain plastic throughout life, they likely retain poised epigenetic elements, and application of single cell epigenetic or multiome tools will be insightful.

Finally, we anticipate that single-cell tools will find applications in clinical diagnostics. Cost concerns may be prohibitive with currently available sequencing modalities, but continued technological advances are expected to reduce sequencing costs in the coming years. The ability of single-cell sequencing to quantify the relative size of diverse cell populations and to identify transcriptional alterations within populations could make it a powerful tool for diagnosing and classifying human disease. For example, despite aganglionosis usually being limited to the distal colon in Hirschsprung disease, there is evidence of altered intestinal motility throughout the gut in mouse models of the disease [66], presumably due to subtle ENS defects. Characterization of the ENS in the ganglionated portion of the bowel in patients with Hirschsprung disease may predict long-term functional outcomes and identify children who may benefit from additional therapy. Single-cell technology thus offers the possibility of personalized medicine for precise diagnosis of pathophysiology and highly tailored individualized therapy.

## 5. Challenges of Single-Cell Analysis in the ENS

In many respects, the challenges of single-cell data acquisition in ENS populations are the same as with any other tissue. Individual cells or nuclei must be obtained from a complex in situ environment with minimal cell loss and without significantly altering the cells’ signaling and transcriptional milieus. The ENS is particularly challenging due to the scarcity of ENS cells within the intestine and the tight apposition of cells within enteric ganglia [22]. A paucity of pan-ENS or neuronal cell surface markers precludes enrichment of many ENS populations by cell sorting, although CD49b has been used to enrich glial populations [64]. Fluorescent reporter systems have facilitated ENS cell enrichment from mouse models [18,25,28], but human ENS cells have remained difficult to obtain. The techniques developed by Drokhlyansky et al. [22] are effective at enriching rare cell populations from human specimens, but they are extremely resource intensive. Improved methods for isolating ENS cells at scale, particularly from human tissue, would be a major advance for the field.

## 6. Conclusions

There has been tremendous growth of single-cell sequencing methods during the past decade. Over the past five years, scRNAseq has been applied extensively to the ENS. Thus far these technologies have predominantly been used to characterize cell populations in normal and diseased states and have revealed significant diversity in many tissues and uncovered previously unknown cell populations. Single-cell methods have already characterized neuronal and glial subpopulations, identified signaling pathways that promote neurogenesis, clarified transcription factors that regulate neuronal function, and suggested how the ENS interacts with non-ENS cells. As sequencing costs decline and tools for single-cell library construction become more widely available, we anticipate that single-cell methods will become a commonplace tool for testing hypotheses and tracking cell behavior in the ENS. Emerging tools for single-cell epigenetics, such as single-nucleus CUT&Tag and single-nucleus ATAC-seq, will soon illuminate mechanisms that maintain or restrict cell fate potential during ENS development and postnatal homeostasis. As tools for robust, high-parameter protein measurement in single-cells mature, we expect that the integration of single-cell transcriptomics, epigenetics, and proteomics will allow scientists to understand how perturbations impact the ENS and ENS-interacting cells on a systems level. These new discoveries have the potential to revolutionize our understanding of disease mechanisms and our development of new treatments.

## Figures and Tables

**Table 1 biomolecules-12-00452-t001:** Summary of Single-Cell Sequencing Studies of the ENS.

Journal	Year	ScRNAseq Platform	Species	Age	Anatomic Region	Ref.
Science	2017	In-house microfluidic system	Mouse	Embryo	Enteric neural crest cells	[15]
Science	2018	inDrop	Zebrafish	Embryo	Whole embryo	[16]
Science	2018	inDrop	Frog	Embryo	Whole embryo	[17]
Cell	2018	10× Genomics	Mouse	P20-30, 6–8-weeks	Brain, spinal cord, small intestine myenteric plexus	[18]
Cell	2018	10× Genomics	Human	Adult	Colon (stroma only)	[19]
Science	2019	10× Genomics	Mouse	E8.5-E10.5	Neural crest cells	[20]
Gastroenterology	2019	Smart-seq, 10× Genomics	Mouse, human	E13.5	Small intestine; iPSCs	[21]
Cell	2020	RAISIN RNA-seq MIRACL-seq	Mouse, human	Adult	Small intestine and colon	[22]
Developmental Cell	2020	10× Genomics	Human	Fetus	Small intestine and colon	[23]
Nature	2020	Drop-seq	Mouse	Adult	Colon (submucosa only)	[24]
Nature Neuroscience	2021	10× Genomics	Mouse	E15.5, E18.5, P21	Small intestine myenteric plexus	[25]
Nature	2021	10× Genomics	Human	Fetus, adult	Small intestine, colon, and mesentery	[26]
			Mouse	10–14 weeks		
Cellular and Molecular Gastroenterology & Hepatology	2021	10× Genomics	Human, mouse	Adult	Colon myenteric plexus	[27]
			Mouse	E17.5	Whole small intestine and colon	
Gastroenterology	2021	10× Genomics	Mouse	6–7.5 weeks	Small intestine and colon	[28]
Cell	2021	10× Genomics	Human	Fetus, adult	Whole small intestine and colon	[29]

## Data Availability

Not applicable.

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
