# Peer review of "Applications of Single-Cell Sequencing Technology to the Enteric Nervous System"

_biomolecules, 2022, doi:10.3390/biom12030452_

Round 1
Reviewer 1 Report
The authors have reviewed the recent papers applying single-cell sequencing tools to the nascent neural crest and developing and mature enteric nervous system (ENS). It is an important area and the applications of scRNAseq platform have accelerated the discovery of the rare cells in ENS and exploration of the developmental trajectories of ENS. They first gave an overview of various single-cell sequencing platforms. The authors then focused on those studies related to ENS and neural crest. Most of the key studies in this area have been included in the review article and were presented in chronological order.
Followings are the major concerns:
- Even the authors have mentioned various single-cell sequencing platforms, they only reviewed the studies of ENS from scRNA-seq. Therefore, the title of the paper should be more precise, “single-cell RNA sequencing” but not “single-cell sequencing” should be used.
- Frist paragraph in Part 3, reference 46, “Using principle component analysis, the authors demonstrated these three groups of cells exist along a continuum between the purely glial/progenitor cells and the purely neuronal cells, and they identified Ret and Phox2b as possible drivers of neuronal differentiation”. It is not clear how the conclusion was made. The authors should elaborate on how scRNAseq data were used to identify the drivers, etc.
- Reference 47 is a study of zebrafish whole embryo from 4phf to 24hpf using scRNA-seq. Even though their data may contain neural crest cells and its derived cells, their relevance to ENS has not been discussed. The authors may need to state how this study is relevant to ENS.
- Similarly, Reference 50 refers to a study of early-stage trunk and cranial neural crest fate commitment, the vagal neural crest cells which is the direct source of ENS has not been discussed. Again, it would be nice to add a sentence to link this finding to ENS study. Alternatively, a new session can be added to include those studies focusing on neural crest.
- The authors may need to elaborate how the model of sequential binary fate decisions published in reference 50 correlates with the results of Morarach et al (reference 51) regarding neuronal differentiation in the embryonic ENS.
- In reference 53, both data of the adult mouse and human ENS at single-cell resolution have been provided. This study covered many interesting analyses. However, the authors only focused on the scRNA-seq technology improvements and clustering. The authors should also discuss the other important findings of the paper (e.g. comparison between human and mouse, cell-cell communication).
- It would be nice to have a table summarizing the key findings from scRNA-seq.
Author Response
We would like to thank the reviewer for their thoughtful comments, which we believe have led to an improved manuscript. Our responses to each comment are as follows:
1) It is true that we have only reviewed papers using scRNA-seq, because to our knowledge there are no papers in the peer-reviewed literature applying other single-cell modalities to the ENS. However, we believe it is important for ENS researchers to be aware of other single-cell tools, and we think it is very likely these tools will be used for ENS research in the near future. We have added a few lines to the introduction (lines 52-56) explaining our reasoning for including all single-cell tools in the review. For these reasons, we favor the current title.
2) We have added text to clarify how Ret was identified as a possible driver of neurogenesis (lines 180-182)
3) Because the ENS is neural crest-derived, we believe studies of early neural crest specification and differentiation are relevant to ENS biology, and ENS scientists should be aware of these studies. We have added text to the introduction (lines 51-52) explaining why these studies are included.
4) Same as #3.
5) We have added text (lines 289-91) clarifying how the cell fate decisions identified in these two papers are similar.
6) We have expanded our discussion of this paper, and included new text discussing some of the analyses the authors performed.
7) A table has been added.
Again, we are grateful to the reviewer for their time and constructive comments. We believe the revised manuscript is stronger.
Reviewer 2 Report
In this manuscript, Guyer et al. reviewed single-cell sequencing technologies and their applications in the enteric nervous system. Single-cell technology has an important role to characterize cell populations in normal and diseased states in ENS, and the rapid development of single-cell fields could provide more and more angles to understand the nature of ENS.
This manuscript is well written and has a clear structure. Overall, it's very useful for readers to understand the basis of single-cell technologies and the current status of single-cell studies in ENS. I suggest biomolecules accept this manuscript after the authors addressed a few minor issues that could further improve the quality of this review.
-
The authors could use a short paragraph to describe the characteristics of ENS samples in the single-cell experiment, and discuss what single-cell platforms have advantages or disadvantages in ENS study. (Like the cell size/cell status/easy to dissociate/etc.)
-
Author Response
We very much appreciate the time this reviewer gave to provide a constructive critique of our manuscript. Our response to the comments are as follows:
1) We have added a section (section 6) discussing the unique challenges of performing single-cell studies on ENS populations. We have also expanded the first paragraph of section 2 (lines 77-82) to emphasize that there is currently no one approach to single-cell sequencing that is regarded as superior to others for ENS applications.
2) A table has been added.